# Multiple Laser Stripe Scanning Profilometry Based on Microelectromechanical Systems Scanning Mirror Projection

**DOI:** 10.3390/mi10010057

**Published:** 2019-01-16

**Authors:** Gailing Hu, Xiang Zhou, Guanliang Zhang, Chunwei Zhang, Dong Li, Gangfeng Wang

**Affiliations:** 1State Key Laboratory for Manufacturing Systems Engineering, Xi’an Jiaotong University, Xi’an 710049, China; hugl@mail.xjtu.edu.cn (G.H.); zgl862203570@stu.xjtu.edu.cn (G.Z.); zcw198811@163.com (C.Z.); ohyailidong@126.com (D.L.); 2School of Mechanical Engineering, Xi’an Jiaotong University, Xi’an 710049, China; 3School of Food Equipment Engineering and Science, Xi’an Jiaotong University, Xi’an 710049, China; 4Key Laboratory of Road Construction Technology and Equipment of MOE, Chang’an University, Xi’an 710064, China; wanggf@chd.edu.cn

**Keywords:** 3D measurement, laser stripe width, vibration noise, MLSSP, MEMS scanning mirror

## Abstract

In traditional laser-based 3D measurement technology, the width of the laser stripe is uncontrollable and uneven. In addition, speckle noise in the image and the noise caused by mechanical movement may reduce the accuracy of the scanning results. This work proposes a new multiple laser stripe scanning profilometry (MLSSP) based on microelectromechanical systems (MEMS) scanning mirror which can project high quality movable laser stripe. It can implement full-field scanning in a short time and does not need to move the measured object or camera. Compared with the traditional laser stripe, the brightness, width and position of the new multiple laser stripes projected by MEMS scanning mirror can be controlled by programming. In addition, the new laser strip can generate high-quality images and the noise caused by mechanical movement is completely eliminated. The experimental results show that the speckle noise is less and the light intensity distribution is more even. Furthermore, the number of pictures needed to be captured is significantly reduced to 1/N (N is the number of multiple laser stripes projected by MEMS scanning mirror) and the measurement efficiency is increased by N times, improving the efficiency and accuracy of 3D measurement.

## 1. Introduction

3D scanning using a structured light projection is widely applied to the measurement of geometric parameters and 3D reconstruction of object surfaces in many fields, including industrial inspection [1,2,3], biomedical treatments [4], culture heritage digitization [5] and food detection [6,7]. This method has several advantages, including noncontact measurement, large measurement ranges, high speed and high accuracy.

Structured light can be divided into two categories: coded-pattern and fixed-pattern light [8]. In coded-pattern light, the most is the Digital Light Processing (DLP) projection, which is widely used in optical measurement because DLP projection is a programmable pattern [9,10,11]. DLP usually projects sinusoidal fringes to obtain the information modulated by the surface of the object, thereby the three-dimensional reconstruction is implemented. This method usually takes a few seconds to implement full-field scanning [12,13,14]. However, the DLP based method is sensitive to measurement environment and the surface reflectivity of the object. The reliability of DLP projection is reduced in the case where the surface reflectivity difference of the measured object is large or the measurement environment is complicated [15]. 

Laser is mainly taken as the light source of a fixed-pattern light [8]. In this laser-based 3D reconstruction system, a laser stripe is projected onto an object and then a camera acquires a series of laser stripe images while the object or the laser stripe is moving forward. The laser stripe is modulated by the shape of the object. Thus, using optical triangulation is possible to calculate 3D information of the object. Due to the intensity and information concentration of the laser stripes, even if the surrounding light is not controlled or the surface of the object to be measured is complex, the measurement results of this method are rarely affected by the environment and the measured object. Therefore, laser-based 3D reconstruction methods can be applied to various industrial environments. Since this method mainly relies on the extraction of the center of the laser strip to obtain the surface information of the measured object, better quality laser stripe can be used to achieve higher measurement accuracy. However, the width of the traditional laser stripe is difficult to reduce and the light intensity distribution is not even, affecting the lateral resolution [16,17]. In addition, the measuring time of this method is mainly subject to the scanning mechanism and the vibration caused by the mechanical movement during the scanning also affects the measurement accuracy [18]. These shortcomings greatly reduce the efficiency and accuracy of measurement results.

In view of the above, it is significant to improve the accuracy of 3D measurement by replacing the traditional line laser scanning with a projector which can project high quality laser stripes. The laser micro-mirror scanning appeared about 40 years ago based on the fact that laser is taken as a light source and micro-mirror as a light modulator [19]. This technology is implemented by microelectromechanical systems (MEMS) manufacturing process which can realize the scanning of laser beam through different control strategies to form a two-dimensional projection image [20,21]. With the development of MEMS, laser micro-mirror scanning technology has found wide applications in engineering in recent years [22,23,24,25]. MEMS scanning mirror has been used in optical coherence tomography (OCT) scans [26,27], time of flight (ToF) cameras [28], 3D confocal scanning microscopes [29,30,31] and other fields of measurement [32,33,34], however, there are few reports of its use in 3D measurement. In this work, a novel 3D measurement method, called multiple laser stripe scanning profilometry (MLSSP) based on MEMS scanning mirror projection, is proposed, which can solve the problems presented above. Compared with conventional laser scanning method, the proposed method has many advantages. The brightness, width, period and position of the new multiple laser stripes projected by MEMS scanning mirror can be controlled by programming, which can generate high-quality measurement images, thereby the MLSSP based on MEMS scanning mirror can improve the accuracy of 3D measurement. The MLSSP is capable of completing full-field scanning measurement in a short time without moving the projector or object, completely eliminates the measurement error caused by the vibration and the measurement efficiency is improved by N times in contrast to the traditional laser-based 3D measurement method. In addition, the proposed method is less affected by industrial environment and surface reflectivity. Due to the robustness, high efficiency and accuracy, this method can be applied to measure objects in various industrial measurement environments including obviously changed surface reflectivity of the measured object and other complicated environment.

This paper is organized as follows: In Section 2, the methods and principles of MLSSP are introduced. In Section 3, experiments on laser stripe performance and 3D measurement are conducted, followed by the discussion of the results and conclusions will be briefly described in Section 4.

## 2. Methods and Principles

In order to improve the scanning speed and reduce the number of pictures in 3D measurement, the MEMS scanning mirror is used to project simultaneously multiple parallel laser stripes. If the number of laser stripes projected is N, the number of images to be captured will be reduced to 1/N and the scan time will also be reduced to 1/N compared to the single laser stripe scanning. The scanning using MLSSP is generally performed in a direction perpendicular to the laser stripe at a fixed interval. By taking the number of laser stripes N=4 as an example, multiple laser stripes scanning process is shown in Figure 1. Controlled by the multiple laser stripe coding method, MEMS scanning mirror projects four parallel laser stripes at time $t_{i}$,$t_{i+1}$,$t_{i+2}$…… with an equal time intervals. Its projection on the continuous surface is in the form of four unoverlapped curves at a certain interval. It only needs to scan the distance of Δx to implement full-field scanning instead of 4Δx by single laser stripe, wherein Δx is the distance of the adjacent laser stipe in the reference surface. 

Multiple laser stripe scanning profilometry involves two key techniques, namely multiple laser stripe coding and 3D reconstruction. This section introduces measurement methods and principles different from traditional laser-based 3D measurement technology.

### 2.1. Multiple Laser Stripe Coding Method

Figure 2 shows the driving signals generating multiple laser stripes projected by MEMS scanning mirror. When the laser incident onto micro-mirror, the micro-mirror fast axis performs simple harmonic motion under the control of a sinusoidal signal to achieve horizontal scanning. The micro-mirror slow axis implements the longitudinal scanning under the control of the sawtooth signal. 

The number, position and output light intensity of the laser stripes are controlled by laser driving pulse signals. There are four equally spaced laser impulse signals in the half cycle of the micro-mirror fast axis driving signal. After reflecting by micro-mirror, each row pixel points of spatial projection field are evenly spaced to generate four laser stripes with equal spacing. 

Step synchronization signal controls the synergy between the laser and fast axis of micro-mirror. After the micro-mirror fast axis completes one cycle of scanning, the slow axis undergoes a slight angular deflection in the vertical direction and achieves accurate step-wise scanning under the control of the sawtooth wave signal, which in turn triggers the fast axis to scan in the next cycle. Figure 3 is one of the multiple laser stripe images projected by MEMS scanning mirror when setting the number of laser stripes N as 4. 

### 2.2. 3D Reconstruction Mechanism

MEMS scanning mirror projects multiple laser stripes onto the object and two cameras capture the deformed laser stripe images. In the experiment, the camera is MV-EM1200M produced by the Microvision of China, with a resolution of 1280 × 960 pixels, pixel size is 3.75 μm × 3.75 μm, equipped with 8 mm fixed lens. The projector is a MEMS scanning mirror driven by a signal control board with two-axis two-dimensional scanning electromagnetic micro-mirror. Figure 4a shows the internal optical path structure of the MEMS scanning mirror. The laser stripe captured by the camera is not a single pixel and hence the centerline of laser stripe needs to be extracted by centerline extraction method, then it is also required to carry out stereo matching and then 3D point cloud images of the object can be obtained. Figure 4b shows a representative schematic diagram of MLSSP. In order to avoid staggered superposition of adjacent laser stripes, the distance between adjacent laser stripes is correspondingly set wider when the depth variation of the measuring object is larger. Taking the left camera as an example, the most suitable distance between adjacent laser stripes is calculated and the number of projected multiple laser stripes will be determined.

The MEMS scanning mirror is placed at the center of the horizontal connection of the two cameras in the binocular system. It is known that the distance between the two cameras is b, the optical distance between the camera’s optical center and the projection device is b/2 and the measurement distance is L. The physical size of the adjacent laser stripe distance is Δx and the maximum step height that can be measured is h. Points A and B are located on the adjacent two laser stripes in the reference plane. According to this triangular relationship, the physical size of the adjacent laser stripe distance is (1)$$\Delta x=\frac{bh}{2L}$$

The resolution of MEMS mirror projection is $\left(u_{p},v_{p}\right)$, the parameter of the projection ratio is $R_{P}$ and the representation of the projection ratio is (2)$$R_{P}=\frac{L}{m}$$
where m is the maximum physical size of the long side of the projected image.

Figure 5 shows the geometry model of the proposed method, by taking the number of laser stripes N=4 as an example. A laser source of MEMS mirror scans to generate multiple laser stripes $A_{1},A_{2},A_{3}$ and $A_{4}$ in the reference plane. One laser stripe is emitted from O, passes through a pixel E in the MEMS mirror plane and falls on a point $A_{1}$ in a reference plane. The adjacent laser stripe passes through a pixel point F in the MEMS mirror plane and falls at a point $A_{2}$ in a reference plane. The length of MN is m with the measurement distance of L, the pixel pitch size EF of the adjacent laser stripe in the MEMS projector is Δp that can be calculated by:(3)$$\Delta p=\frac{u_{p}}{m}\Delta x=\frac{u_{p}R_{p}bh}{2L^{2}}$$

In this way, if the MEMS scanning mirror projects N laser stripes and there are N laser stripe information in each line of the pictures captured by the left and right cameras, these laser strips in left and right images are matched in order. However, the laser stripe may be missing or broken when measuring the edge of the object, as shown in Figure 6. In this case, if the stereo matching is performed according to the principle of simple matching order, the matching of the laser stripes are difficult and results may be wrong. In order to solve this problem, the fixed area needs to be divided for each laser stripe.

When the surface depths of the measurement object are different, each laser stripe has a certain range in the image captured by camera. So the multiple laser stripe image can be divided into N areas. The ith laser stripe is always fixed in the ith area, then the left and right images can be matched according to the corresponding laser stripe area.

The following section describes the method of dividing laser stripes area by taking the number of laser stripes N = 4 as an example. Firstly, place the white flat plate parallel to two optical centers of the cameras with a measurement distance of L and then capture the laser stripe image by the camera. Calculate the longitudinal pixel coordinates of four laser stripe centerlines in the image are $xn_{1},xn_{2},xn_{3}$ and $xn_{4}$. Secondly, translate the white calibration plate to the position with the measurement distance of L–h, capture the laser stripe image, calculate the longitudinal pixel coordinates of the four laser stripe centerline in the distance of L–h as $xf_{1},xf_{2},xf_{3}$ and $xf_{4}$. Then the fixed area of each laser stripe in the image respectively are $\left(xn_{1},xf_{1}),(xn_{2},xf_{2}),(xn_{3},xf_{3}\right)$ and $\left(xn_{4},xf_{4}\right)$ in this position. Then it can be linearly shifted by one or two pixels as unit step in the projected pixel plane. The methods of dividing area for each laser stripe of left and right images are the same. After dividing all the laser stripes area of left and right images, stereo matching and reconstruction work can be performed within the fixed areas.

## 3. Experiment and Analysis

### 3.1. Robust Test

In order to verify the robustness of the laser stripe projected by MEMS scanning mirror, the laser stripe images captured by cameras are investigated. Firstly, two laser stripes, one is projected by traditional linear laser device and another by a MEMS mirror, are simultaneously projected onto a flat surface. The captured image is shown in Figure 7a, wherein the left side is a traditional laser stripe and the right side is a MEMS projection laser stripe. The pixels in the up, middle and bottom cross-sections of the captured image are drawn in Figure 7b. It can be seen that the traditional laser stripe image has more speckle noise and the width and intensity distribution of laser stripe are not even, while the laser stripe projected by MEMS mirror is more even and finer, which is more helpful to improve the measure accuracy. 

In addition, in order to test the depth of field of the MEMS mirror, the MEMS and DLP project the same size spot. The camera is aligned with the light exit of the MEMS scanning mirror and the DLP. By changing the distance between the camera and the light exit, the relationship between the spot size and the projection distance is tested. To prevent interference from other ambient light, the experiment is conducted in the darkroom. The size of the MEMS mirror spot and the size of the DLP spot at different projection distances are shown in Figure 8. The solid dot is the measurement value of the spot diameter and the curve is the relationship between the fitted spot size and the projection distance.

It can be seen from the experiment that the spot projected by MEMS mirror is focused at 500 mm after passing through the lens, the spot size is about 300 μm and the spot diameter is kept within 0.5 mm within the 300 mm from the focus point. The red curve is the depth of field simulation of MEMS scanning mirror. The spot projected by DLP is also focused at 500 mm and the spot quickly blurs after deviating from the focus position. The blue curve is the depth of field simulation of DLP technology. The experiment results show that the spot size of the MEMS scanning mirror is smaller than DLP at the focus position and the spot diameter changes little within the measurement area. This experiment verifies the advantages of the MEMS scanning mirror with a large depth of field. In contrast, the DLP projection has a smaller depth of field and is sensitive to different reflectivity of object surface and to ambient light, which greatly reduce the accuracy of results. Thus DLP could not replace laser when using line structured light method in a complex environment. Therefore, DLP is generally used only as a fringe projection for full-field scanning measurement and the scanning time is a few seconds. The above experiments proved that the laser stripes projected by MEMS scanning mirror are robust and more suitable for line structured light measurement.

### 3.2. Reconstruction of A Face Plaster Model

A human-face plaster model was reconstructed by the proposed method and traditional laser-based 3D reconstruction method. The resolution of the MEMS scanning mirror is 1280 × 720 pixels. If the object is scanned pixel by pixel in the horizontal direction with a single laser stripe, the number of the laser stripe images needed to be captured is 1280. According to Equation (3), the most minimum pixel pitch for MEMS scanning mirror can be calculated as 285 pixels. Since the number of projected laser stripes must be an integer, the number of laser stripes is selected to be 4 for scanning and reconstruction. In this situation, only 320 images can implement the measurement of the face plaster model. The frame rate of the camera is 30 frames/s and the frame rate of the MEMS scanning mirror is 50 frames/s. The maximum frame rate of the system 30 frames/s is selected and the time taken for the field measurement is only 10.6 s. Then the laser stripe centerline is extracted by the gravity center extraction method [35] and the three-dimensional reconstruction of the model is implemented by binocular stereo matching method based on a fixed area.

Figure 9 shows the results of reconstruction by traditional laser scanning method and the MLSSP. Due to the large amount of speckle noise and the unevenness of the brightness of the conventional laser stripe, the images captured by cameras need filtering and removing noises before 3D reconstruction. Otherwise, a large number of mismatched points will appear in the stereo matching, which can reduce the accuracy of 3D reconstruction. Figure 9a shows the result of three-dimensional reconstruction by the conventional laser stipe scanning method after noise reduction. The texture features of the mouth and eyes are smoothed and errors may occur on the left face. For the MEMS scanning mirror, no noise reduction is required before 3D reconstruction. More importantly, no mechanical motion device is needed and it only takes 15 s to implement the scanning and 3D reconstruction. Compared with traditional methods, the measurement time by MLSSP is greatly reduced. Figure 9b is the 3D reconstruction model by MLSSP. The facial model is reconstructed very clearly and the surface curvature changes such as the nose and eyes are also clearly reconstructed. It can be seen that the 3D reconstruction model has higher accuracy by MLSSP contrasting to traditional laser-based 3D reconstruction method.

### 3.3. Accuracy Test

The precision of the system should be judged by measuring the standard objects with known size. As shown in Figure 10a two gauge blocks are stacked to form a standard step. The upper ceramic gauge block is 5 mm thick with tolerance of ±6 μm. The three-dimensional model of gauge block above is reconstructed by the MLSSP as shown in Figure 10b. The red plane is the result of the reconstruction of the ceramic gauge above and the blue is the result of the reconstruction of the ceramic gauge below. The reconstructed profiles appear to be correct. To further check the measurement accuracy, the cross section of the recovered profiles with x=30mm is plotted in Figure 10c. It is detected that the step height is around 5 mm. The measurement size of the gauge block above is subtracted by the true size of 5 mm to obtain an error distribution of all points as shown in Figure 11. The different colors represent the different errors of the upper surface of the standard gauge block. The maximum residual of the step size can be obtained as (/mm) (4)$$e_{\max}=\max(h_{i}-h_{real})$$
where $h_{i}$ is the measurement size of the ith pixel and $h_{real}$ is the true size of 5 mm. If $e_{i}$ is the measurement deviation of the ith pixel and $\bar{e}$ is the value of average deviation, the standard deviation is calculated by the following equation (/mm) (5)$$\sigma =\sqrt{\frac{1}{N}\sum_{i=1}^{N}{\left(e_{i}-\bar{e}\right)}^{2}}$$

According to measurement error distribution of standard gauge block upper surface, the result of $e_{\max}$ is 0.1428 mm and the standard deviation of σ is 0.0535 mm.

The image processing methods in the experiment use the gravity center extraction method [35] and linear interpolation stereo matching. However, the methods are not focal of my research, which can be replaced by other more accurate method to improve the accuracy of 3D reconstruction. In addition, the accuracy of the method is also affected by the surface accuracy of the ceramic block itself, the systematic errors and the external disturbances, which can be further reduced during the experiment. It can be seen that the measurement system of the MLSSP has excellent three-dimensional measurement performance and can provide powerful technological support for 3D inspection, reverse engineering and rapid manufacturing represented by 3D printing.

## 4. Conclusions

In summary, a robust multiple laser stripe scanning method for three-dimensional measurement is proposed in this work. It is based on programmable projection and fast scanning of MEMS scanning mirror. The test results show that the laser stripes projected by MEMS scanning mirror are more even and finer than traditional laser stripe. MEMS scanning mirror is more suitable for the line-structured light measurement than DLP and traditional laser. In addition, the number, period and scanning direction of multiple laser stripes projected by MEMS scanning mirror can be adjusted by calculating and programming. The MLSSP can implement the automatic scanning without the need of mechanical motion devices, which eliminates vibration noise caused by mechanical motion. Moreover, the number of the pictures needed to be captured is reduced to 1/N. It greatly improves the efficiency of online 3D measurement. Finally, the reliability tests are conducted, including reconstruction of a face plaster model and accuracy test. The human-face plaster model is reconstructed clearly including the surface curvature changes of mouth and eyes. Accuracy test shows that standard deviation is 0.0535 mm and the accuracy of 3D reconstruction can be further improved. 

## Figures and Tables

**Figure 1 micromachines-10-00057-f001:**
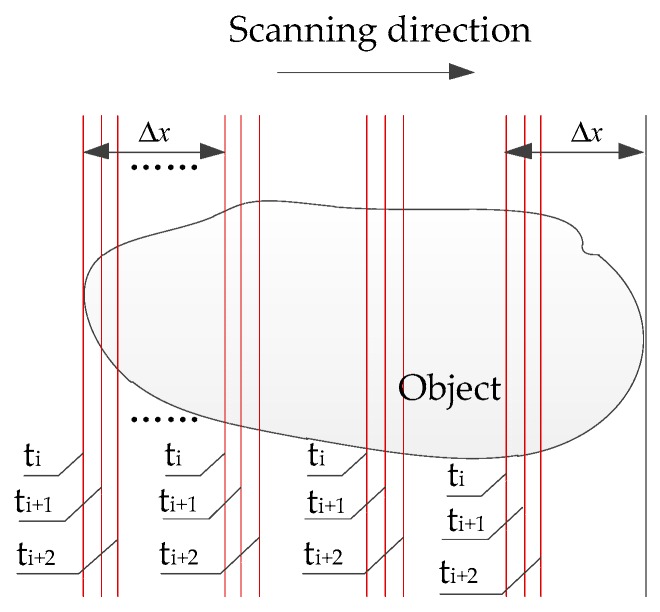
Scanning process of multiple laser stripe scanning profilometry (MLSSP) taking the number of laser stripes of N=4.

**Figure 2 micromachines-10-00057-f002:**
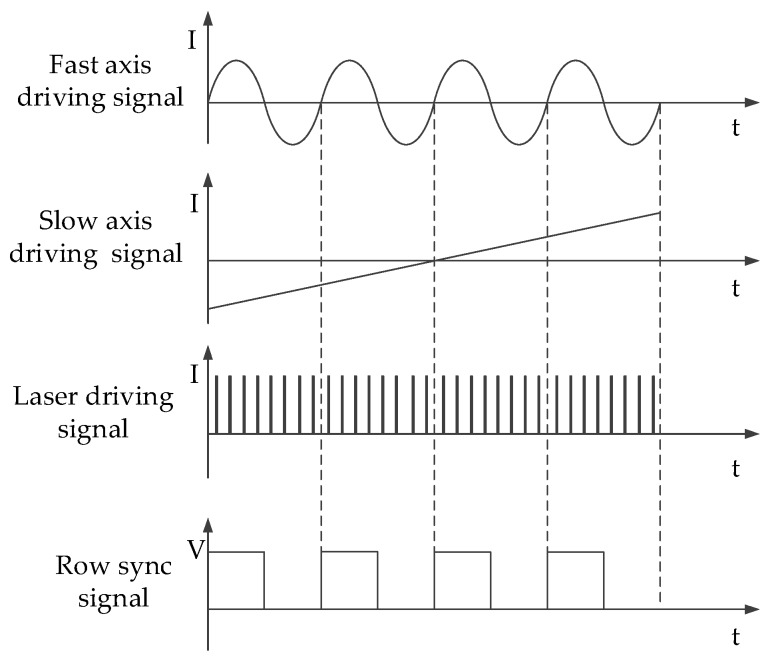
Driving signals generating multiple laser stripes projected by microelectrochemical systems (MEMS) scanning mirror.

**Figure 3 micromachines-10-00057-f003:**
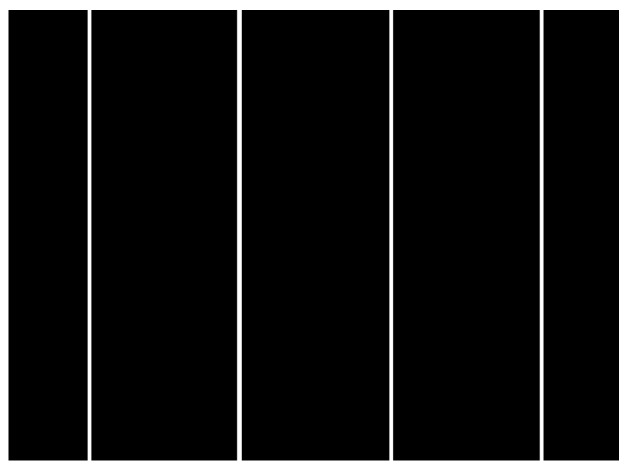
A four-laser stripe image projected by MEMS scanning mirror.

**Figure 4 micromachines-10-00057-f004:**
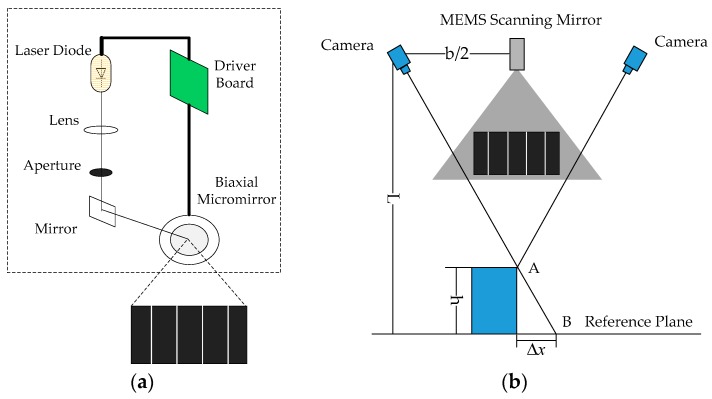
(**a**) Internal structure of the MEMS scanning mirror. (**b**) Schematic diagram of MLSSP.

**Figure 5 micromachines-10-00057-f005:**
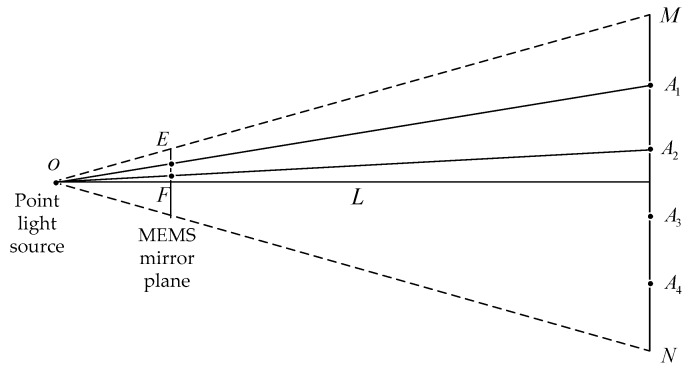
Geometry model of MLSSP.

**Figure 6 micromachines-10-00057-f006:**
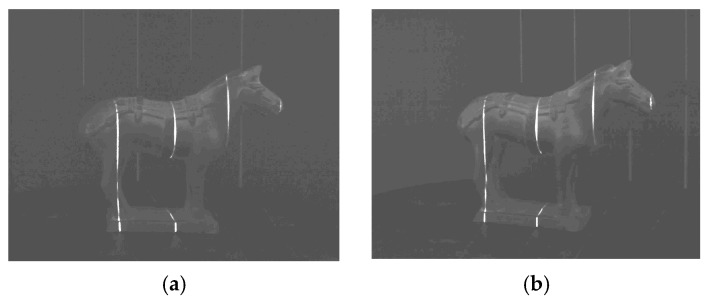
Missing and damaged laser stripes: (**a**) Left image and (**b**) Right image.

**Figure 7 micromachines-10-00057-f007:**
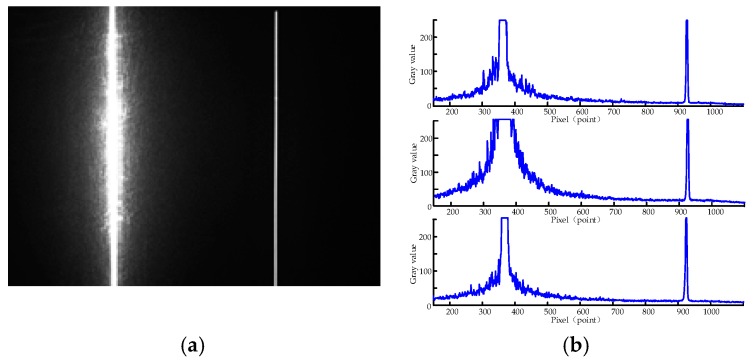
Laser stripe contrast: (**a**) Two laser stripes captured in the same field of view; and (**b**) Gray distribution of different rows of pixels.

**Figure 8 micromachines-10-00057-f008:**
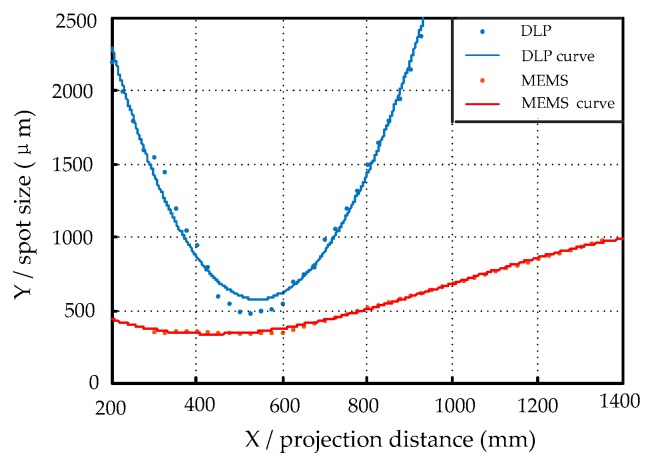
Depth of field comparison between Digital Light Processing (DLP) and MEMS scanning mirror.

**Figure 9 micromachines-10-00057-f009:**
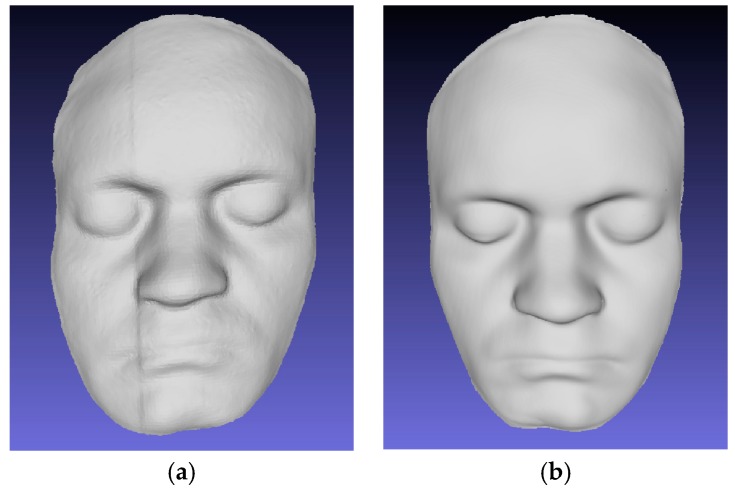
Three Dimensional reconstruction results contrast: (**a**) Traditional laser scanning method; and (**b**) MLSSP.

**Figure 10 micromachines-10-00057-f010:**
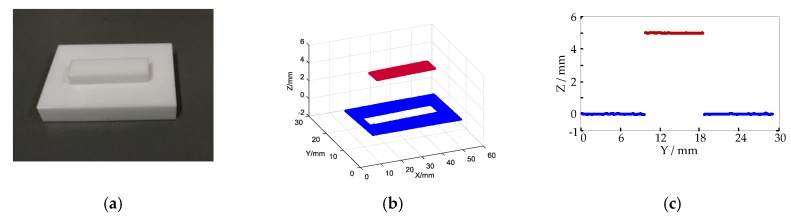
(**a**) Standard gauge block placed on a standard plane; (**b**) Three-dimensional model; (**c**) Cross section of the reconstructed 3D profile with x=30mm.

**Figure 11 micromachines-10-00057-f011:**
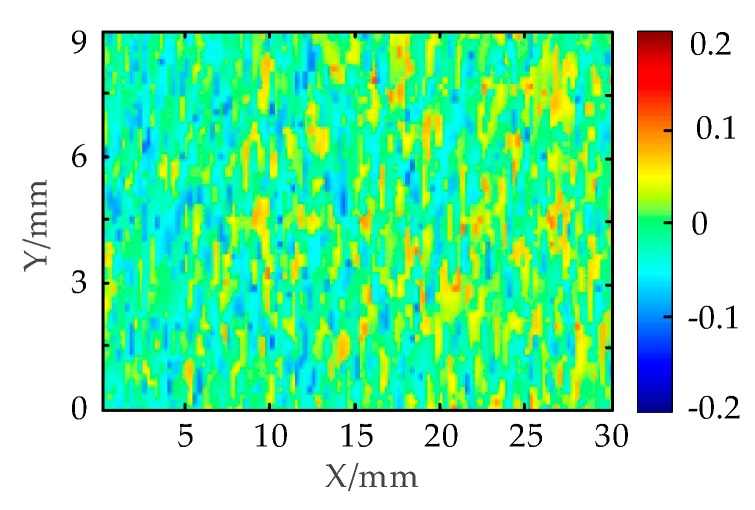
Measurement error distribution of standard gauge block upper surface.

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
