# Peer review of "Multiple Laser Stripe Scanning Profilometry Based on Microelectromechanical Systems Scanning Mirror Projection"

_micromachines, 2019, doi:10.3390/mi10010057_

Round 1
Reviewer 1 Report
In this paper, the authors introduced a traditionally laser-based 3D measurement technology; the width of the laser stripe is uncontrollable and uneven. In addition, speckle noise in the image and noise caused by mechanical movement reduce the accuracy of the scanning results. This work proposes a new technological multiple laser stripe scanning profilometry (MLSSP) characterized with microelectromechanical systems (MEMS) scanning mirror which can project the required laser stripe. It can implement full-field scanning in a short time, which does not need to move the measured object or camera. The idea behind this is interesting. However, I still have quite a number of concerns in this manuscript. There are times where there are not enough data to support the conclusions of the author. Please see some of the major concerns below.
1.The information of the MLSSP setup is not enough (figure 1). The authors should give much more information about this. So the readers can get its reproducibility.
2. The authors should give much more information about the novelty of this paper, especially the effect of using the MLSSP, which applications can used it?
3. The tolerance analysis, which can offer a good guide for the fabrication requirement, and the key parameters, need to be added in the results section. I think the laser scan along x,y,z axis can be show the analysis.
4. More references need to be included in the introduction part to understand the applications of using high laser applications:
1. "Prospects for diode-pumped alkali-atom-based hollow-core photonic-crystal fiber lasers".Optics Letters, 39(16), 2014 (4655-4658)
2. "Super-resolved Raman spectra of Toluene and Toluene-chlorobenzene mixture"
Spectroscopy Letters, 48(6), 2015 (431-435)
Author Response
Please find the responses in the attachment.

Reviewer 2 Report
This manuscript presents a new technological multiple laser stripe scanning profilometry (MLSSP),that uses MEMS mirror to project the required laser stripe and enable fast full-field scanning without moving the object or camera. This work is good. The manuscript can be accepted after some minor revision. Here are some comments for authors to consider:
Please provide the full-field scanning speed of DLP based method. It will be helpful for readers to understand the proposed MEMS Mirror based MLSSP method.
Please check Line 98-99 _"The projector is SONY MP-CL1 MEMS mirror." and Fig.4 _MEMS scanning mirror. In this work, is the MEMS mirror of SONY MP-CL1 projector or the projector was used to build the MLSSP? If only MEMS mirror was used in this instrument, the laser and optical parts should be added in Fig. 4.
Author Response

(The authors gave the same response as above.)
